# CONTEXTUAL TEXT STYLE TRANSFER

## ABSTRACT

In this paper, we introduce a new task, Contextual Text Style Transfer, to translate a sentence within a paragraph context into the desired style (e.g., informal to formal, offensive to non-offensive). Two new datasets, Enron-Context and Reddit-Context, are introduced for this new task, focusing on formality and offensiveness, respectively. Two key challenges exist in contextual text style transfer: 1) how to preserve the semantic meaning of the target sentence and its consistency with the surrounding context when generating an alternative sentence with a specific style; 2) how to deal with the lack of labeled parallel data. To address these challenges, we propose a Context-Aware Style Transfer (CAST) model, which leverages both parallel and non-parallel data for joint model training. For parallel training data, CAST uses two separate encoders to encode each input sentence and its surrounding context, respectively. The encoded feature vector, together with the target style information, are then used to generate the target sentence. A classifier is further used to ensure contextual consistency of the generated sentence. In order to leverage massive non-parallel corpus and to enhance sentence encoder and decoder training, additional self-reconstruction and back-translation losses are introduced. Experimental results on Enron-Context and Reddit-Context demonstrate the effectiveness of the proposed model over state-of-the-art style transfer methods, across style accuracy, content preservation, and contextual consistency metrics.[1]

## 1 INTRODUCTION

Text style transfer has recently been applied to many applications with remarkable success (e.g., sentiment manipulation, formalized writing). Early work relied on parallel corpora with a sequence-to-sequence learning framework (Bahdanau et al., 2015; Jhamtani et al., 2017). However, collecting annotations for parallel data is highly time-consuming. There has been a recent surge of interest in developing text style transfer models using non-parallel data (Hu et al., 2017; Li et al., 2018; Prabhumoye et al., 2018; Subramanian et al., 2018), assuming that disentangling style information from semantic content can be achieved in an auto-encoding fashion with the introduction of additional regularizers (e.g., adversarial discriminators (Shen et al., 2017) or language models (Yang et al., 2018)).

Despite promising results, these techniques still have a long way towards practical use. Specifically, existing models mostly focus on sentence-level rewriting. However, in real-world applications, sentences typically reside in a proper context such as a paragraph. For example, in the formalized writing task, the rewritten span should align well with the surrounding context (e.g., personal email, scientific content) to keep a coherent text flow. Taking a single sentence as the sole input of a style transfer model may fail to preserve topical coherency of the generated sentence with its surrounding context, resulting in poor semantic and logical consistency on the paragraph level (see Example C in Table 4).

Motivated by this, we propose and investigate a new task - *Contextual Text Style Transfer*. Given a paragraph, the system aims to automatically edit sentences into a desired style, while keeping the edited section topically coherent with its surrounding context. To achieve this goal, we propose a novel Context-Aware Style Transfer (CAST) model, by jointly considering style transfer and context alignment. For parallel training data, CAST uses two separate encoders to encode the source sentence and its surrounding context, respectively, and a decoder to translate the encoded features

---

[1]Source code, and collected new datasets will be released upon acceptance.

into the target sentence. A pre-trained coherence classifier is further applied to regularize the generated target sentence to be consistent with the context. To overcome the data sparsity issue, we further leverage non-parallel data by using a hybrid approach. With large-scale non-parallel corpus, the training of the sentence encoder and decoder are enhanced via additional self-reconstruction and back-translation objectives. A pre-trained style classifier is also used for style regularization. The final CAST model is jointly trained with both parallel and non-parallel data.

As this is a newly proposed task, we also introduce two new datasets, *Enron-Context* and *Reddit-Context*, collected via crowdsourcing. The former contains 14,734 formal vs. informal paired samples from Enron (Klimt & Yang, 2004) (an email dataset), and the latter contains 23,158 offensive vs. non-offensive paired samples from Reddit (Serban et al., 2017). Each paired sample contains an original sentence and a human-rewritten sentence with the desired style, accompanied by its paragraph context. Besides this, in order to enhance model training, we exploit additional 28,375/29,774 formal/informal non-parallel sentences from GYAFC (Rao & Tetreault, 2018), and 53,028/53,714 offensive/non-offensive non-parallel sentences from Reddit (dos Santos et al., 2018).

The main contributions of this work are summarized as follows: ($i$) We propose a new task - Contextual Text Style Transfer, which aims to translate an input sentence into a desired style, while preserving its style-irrelevant semantics and topical consistency with the surrounding context. ($ii$) We introduce two new datasets for this task, Enron-Context and Reddit-Context, which provide reliable benchmarks for measuring contextual style transfer models. ($iii$) We present a new model - Context-Aware Style Transfer (CAST), which jointly optimizes the generation quality of the target sentence and its topical coherency with adjacent sentences. Extensive experiments on these two new datasets demonstrate that the proposed CAST model outperforms state-of-the-art baselines.

## 2 RELATED WORK

**Text Style Transfer** Text style transfer aims to modify an input sentence into a desired style while preserving its style-independent semantics. Previous work has explored this as a sequence-to-sequence learning task using parallel corpora with paired source/target sentences in different styles. For example, Jhamtani et al. (2017) pre-trained word embeddings by leveraging external dictionaries mapping Shakespearean words to modern English words and additional text. However, available parallel data in different styles are very limited. Therefore, there is a recent surge of interest in considering a more realistic setting, where only non-parallel stylized corpora are available. A typical approach is: ($i$) disentangling latent space as content and style features; then ($ii$) generating stylistic sentences by tweaking style-relevant features and passing them through a decoder, together with the original content-relevant features (Xu et al., 2018).

Many of these approaches borrowed the idea of adversarial discriminator/classifier from the Generative Adversarial Network (GAN) framework (Goodfellow et al., 2014). For example, Shen et al. (2017); Fu et al. (2018); Lample et al. (2018) used adversarial classifiers to force the decoder to transfer the encoded source sentence into a different style/language. Alternatively, Li et al. (2018) achieved disentanglement by filtering stylistic words of input sentences. Another direction for text style transfer without parallel data is using back-translation (Prabhumoye et al., 2018) with a denoising auto-encoding objective (Logeswaran et al., 2018; Subramanian et al., 2018).

Regarding the tasks, sentiment transfer is one of the most widely studied problems. From informality to formality (Rao & Tetreault, 2018) is another direction of text style transfer, aiming to change the style of a given sentence to more formal text. dos Santos et al. (2018) presented an approach to transferring offensive text to non-offensive based on social network data. In Prabhumoye et al. (2018), the authors proposed the political slant transfer task. However, all these previous studies did not directly consider context-aware text style transfer, which is the main focus of this work.

**Context-aware Text Generation** Our work is related to context-aware text generation (Mikolov & Zweig, 2012; Tang et al., 2016), which can be applied to many NLP tasks (Mangrulkar et al., 2018). For example, previous work has investigated language modeling with context information (Wang & Cho, 2015; Wang et al., 2017), treating the preceding sentences as context. There are also studies on response generation for conversational systems (Sordoni et al., 2015b; Wen et al., 2015), where dialogue history is treated as a context. Zang & Wan (2017) introduced a neural model to generate long reviews from aspect-sentiment scores given the topics. Vinyals & Le (2015) proposed a model

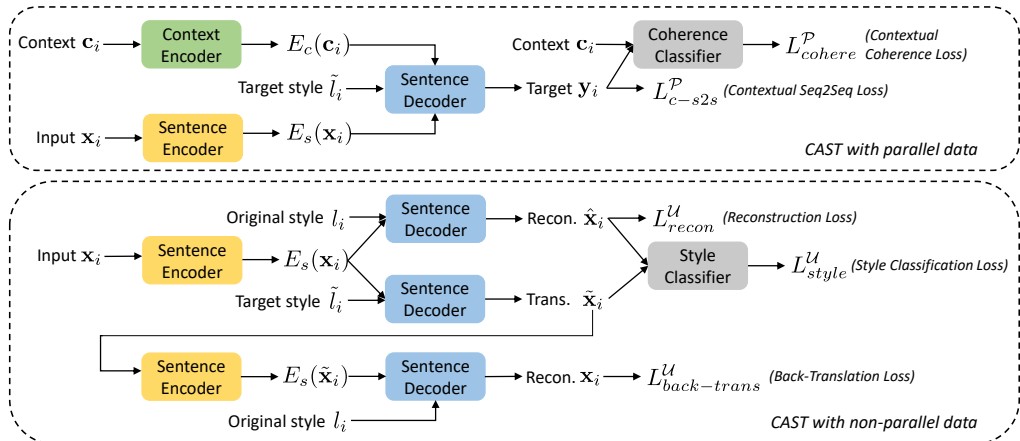

Figure 1: Model architecture of the proposed CAST model for contextual text style transfer. Both the training paths share the same sentence encoder and decoder. See Sec. 3 for details.

to predict the next sentence given the previous sentences in a dialogue session. Sordoni et al. (2015a) presented a hierarchical recurrent encoder-decoder model to encode dialogue context. Our work is the first to explore context information in the text style transfer task.

## 3 CONTEXTUAL TEXT STYLE TRANSFER

In this section, we first describe the problem definition and provide an overview of the model architecture in Section 3.1. Section 3.2 presents the proposed Context-Aware Style Transfer (CAST) model with parallel data, and Section 3.3 further introduces how to augment the CAST model with non-parallel data in a hybrid approach.

### 3.1 OVERVIEW

**Problem Definition**    The problem of contextual text style transfer is defined as follows. Given a style-labelled parallel dataset $\mathcal{P} = \{(\mathbf{x}_i, l_i), (\mathbf{y}_i, \tilde{l}_i), \mathbf{c}_i\}_{i=1}^{M}$, where the $i$-th instance contains the original sentence $\mathbf{x}_i$ in style $l_i$, its corresponding rewritten sentence $\mathbf{y}_i$ in another style $\tilde{l}_i$, and the paragraph context $\mathbf{c}_i$. $\mathbf{x}_i$ and $\mathbf{y}_i$ are expected to contain the same semantic content, but in different language styles (i.e., $l_i \neq \tilde{l}_i$). The goal is to transform $\mathbf{x}_i$ in style $l_i$ to $\mathbf{y}_i$ in style $\tilde{l}_i$, while keeping the sentence $\mathbf{y}_i$ semantically coherent with its context $\mathbf{c}_i$. In practice, labelled parallel data may be difficult to garner. Therefore, we assume that additional non-parallel data $\mathcal{U} = \{(\mathbf{x}_i, l_i)\}_{i=1}^{N}$ can be leveraged to enhance overall model training.

**Training Objective**    The overall architecture of the proposed CAST model is illustrated in Figure 1. The hybrid model training process consists of two paths, one for parallel data and the other for non-parallel data. In the parallel path, a Seq2Seq loss and a contextual coherence loss are defined, to learn the two encoders (sentence encoder and context encoder) and the sentence decoder with labeled parallel data. The non-parallel path is designed to further enhance the sentence encoder and decoder with three additional losses: ($i$) a self-reconstruction loss; ($ii$) a back-translation loss; and ($iii$) a style classification loss. The overall training objective, taking both parallel and non-parallel paths into consideration, can be written as:

$$L_{final}^{\mathcal{P},\mathcal{U}} = L_{c-s2s}^{\mathcal{P}} + \lambda_1 L_{cohere}^{\mathcal{P}} + \lambda_2 L_{recon}^{\mathcal{U}} + \lambda_3 L_{back-trans}^{\mathcal{U}} + \lambda_4 L_{style}^{\mathcal{U}}, \quad (1)$$

where $\lambda_1$, $\lambda_2$, $\lambda_3$ and $\lambda_4$ are hyper-parameters to balance different objectives. Each of these loss terms will be explained in the following sub-sections.

### 3.2 CAST WITH PARALLEL DATA

In this subsection, we discuss the training objective associated with parallel data, consisting of ($i$) a contextual Seq2Seq loss; and ($ii$) a contextual coherence loss.

**Contextual Seq2Seq Loss**  When parallel data are available, a Seq2Seq model can be directly learned for text style transfer. We denote Seq2Seq model as $(E, D)$, where the semantic representation of sentence $\mathbf{x}_i$ is extracted by the encoder $E$ (i.e., $E(\mathbf{x}_i)$), and the decoder $D$ aims to learn a conditional distribution of $\mathbf{y}_i$ given the encoded feature $E(\mathbf{x}_i)$ and style $\tilde{l}_i$:

$$L_{s2s}^{\mathcal{P}} = - \mathbb{E}_{\mathbf{x}_i, \mathbf{y}_i \sim \mathcal{P}} \log p_D(\mathbf{y}_i | E(\mathbf{x}_i), \tilde{l}_i) . \tag{2}$$

However, in such a sentence-to-sentence style transfer setting, the context of the paragraph is ignored, which if well utilized, could help improve the quality of generated text (such as paragraph-level topical coherence).

Thus, to take advantage of the paragraph context $\mathbf{c}_i$ information, we use two separate encoders $E_s$ and $E_c$ to encode the sentence and the context independently. The outputs of the two encoders are combined via a linear layer, to obtain a context-aware sentence representation, which is used for generating the target sentence. The model is trained to minimize the following loss:

$$L_{c-s2s}^{\mathcal{P}} = - \mathbb{E}_{\mathbf{x}_i, \mathbf{c}_i, \mathbf{y}_i \sim \mathcal{P}} \log p_D(\mathbf{y}_i | E_s(\mathbf{x}_i), E_c(\mathbf{c}_i), \tilde{l}_i) . \tag{3}$$

Compared with Eqn. (2), the use of $E_c(\mathbf{c}_i)$ makes the text style transfer process context-dependent. The generated sentence can be denoted as $\tilde{\mathbf{y}}_i = D(E_s(\mathbf{x}_i), E_c(\mathbf{c}_i), \tilde{l}_i)$.

**Contextual Coherence Loss**  To enforce contextual coherence (i.e., to make the generated sentence $\mathbf{y}_i$ align with the surrounding context $\mathbf{c}_i$), we train a coherence classifier that aims to distinguish whether $\mathbf{c}_i$ is the context of $\mathbf{y}_i$, by adopting a language model with an objective similar to next sentence prediction (Devlin et al., 2019).

Specifically, assume that $\mathbf{y}_i$ is the $t$-th sentence of a paragraph $\mathbf{p}_i$ (i.e., $\mathbf{y}_i = \mathbf{p}_i^{(t)}$), and $\mathbf{c}_i = \{\mathbf{p}_i^{(0)}, \ldots, \mathbf{p}_i^{(t-1)}, \mathbf{p}_i^{(t+1)}, \ldots, \mathbf{p}_i^{(T)}\}$ is its surrounding context. We first reconstruct the paragraph $\mathbf{p}_i = \{\mathbf{p}_i^{(0)}, \ldots, \mathbf{p}_i^{(T)}\}$ by inserting $\mathbf{y}_i$ into the proper position in $\mathbf{c}_i$, denoted as $[\mathbf{c}_i; \mathbf{y}_i]$. Based on this, we obtain a paragraph representation $\mathbf{u}_i$ via a language model encoder. Then, we apply a linear layer to the representation, followed by a $\tanh$ function and a softmax layer to predict a binary label $s_i$, which indicates whether $\mathbf{c}_i$ is the context of $\mathbf{y}_i$ :

$$\mathbf{u}_i = \mathrm{LM}([\mathbf{c}_i; f(\mathbf{y}_i)]) \tag{4}$$

$$p_{\mathrm{LM}}(s_i | \mathbf{c}_i, \mathbf{y}_i) = \mathrm{softmax}\left(\tanh\left(\mathbf{W}\mathbf{u}_i + \mathbf{b}\right)\right) . \tag{5}$$

where LM represents the language model encoder, and $s_i = 1$ indicates that $\mathbf{c}_i$ is the context of $\mathbf{y}_i$. Note that since $\tilde{\mathbf{y}}_i$ are discrete tokens which are non-differentiable, we use the continuous feature, denoted as $f(\tilde{\mathbf{y}}_i)$, that generates $\tilde{\mathbf{y}}_i$ as the input of the language model. We construct paired data $\{\mathbf{y}_i, \mathbf{c}_i, s_i\}_{i=1}^N$ for training the classifier, where the negative samples are generated by replacing a sentence in a paragraph with another random sentence. After pre-training, the coherence classifier is used to obtain the contextual coherence loss:

$$L_{cohere}^{\mathcal{P}} = - \mathbb{E}_{\mathbf{x}_i, \mathbf{c}_i \sim \mathcal{P}} \log p_{\mathrm{LM}}(s_i = 1 | \mathbf{c}_i, f(\tilde{\mathbf{y}}_i)) . \tag{6}$$

Intuitively, minimizing $L_{cohere}^{\mathcal{P}}$ encourages the $\tilde{\mathbf{y}}_i$ to blend better to its context $\mathbf{c}_i$. Note that the coherence classifier is pre-trained, and fixed during the training of the CAST model. The above coherence loss can be used to update the parameters of $E_s, E_c$ and $D$ during model training.

## 3.3 CAST with Non-parallel Data

For the contextual style transfer task, there are not many parallel datasets available with style-labeled paragraph pairs. To overcome the data sparsity issue, we propose to further boost the CAST model by leveraging additional non-parallel data $\mathcal{U} = \{(\mathbf{x}_i, l_i)\}_{i=1}^N$, which are less expensive to collect. In order to fully exploit $\mathcal{U}$ to enhance the training of the sentence encoder and decoder $(E_s, D)$, we introduce three additional training losses, detailed below.

**Reconstruction Loss**  The reconstruction loss aims to encourage $E_s$ and $D$ to reconstruct the input sentence itself, if the desired style is the same as the input. The corresponding objective is similar to Eqn. (2):

$$L_{recon}^{\mathcal{U}} = - \mathbb{E}_{\mathbf{x}_i \sim \mathcal{U}} \log p_D(\mathbf{x}_i | E_s(\mathbf{x}_i), l_i) . \tag{7}$$

| Dataset | # sent. | # rewritten sent. | # words per sent. | # words per paragraph | # vocabulary |
|---|---|---|---|---|---|
| Reddit-Context | 14,734 | 14,734 | 9.4 | 38.5 | 4,622 |
| Enron-Context | 23,158 | 25,259 | 7.6 | 25.9 | 2,196 |

Table 1: Statistics on Enron-Context and Reddit-Context datasets.

Compared with Eqn. (2), here we encourage the decoder $D$ to recover $\mathbf{x}_i$'s original stylistic property as accurate as possible when given the style label $l_i$. The self-reconstructed sentence is denoted as $\hat{\mathbf{x}}_i = D(E_s(\mathbf{x}_i), l_i)$.

**Back-Translation Loss** The back-translation loss requires the model to reconstruct the input sentence after a transformation loop. Specifically, the input sentence $\mathbf{x}_i$ is first transferred into the target style, i.e., $\tilde{\mathbf{x}}_i = D(E_s(\mathbf{x}_i), \tilde{l}_i)$. Then the generated target sentence is transferred back into its original style, i.e., $\hat{\mathbf{x}}_i = D(E_s(\tilde{\mathbf{x}}_i), l_i)$. The back-translation loss is defined as:

$$L^{\mathcal{U}}_{back-trans} = - \mathop{\mathbb{E}}_{\mathbf{x}_i \sim \mathcal{U}, \tilde{\mathbf{x}}_i \sim p_D(\mathbf{y}_i|E_s(\mathbf{x}_i), \tilde{l}_i))} \log p_D(\mathbf{x}_i|E_s(\tilde{\mathbf{x}}_i), l_i). \tag{8}$$

where the source style is denoted as $l_i$, and the target style is denoted as $\tilde{l}_i$.

**Style Classification Loss** To further boost the model, we use $\mathcal{U}$ to train a classifier to predict the style of a given sentence, and regularize the training of $(E_s, D)$ with the pre-trained style classifier. Specifically, the objective for training the style classifier is:

$$L_{style} = - \mathop{\mathbb{E}}_{\mathbf{x}_i \sim \mathcal{U}} \log p_C(l_i|\mathbf{x}_i). \tag{9}$$

where $p_C(\cdot)$ denotes the style classifier. After the classifier is trained, we keep its parameters fixed, and apply it to update the parameters of $(E_s, D)$. Specifically, the style classification loss defined over the pre-trained style classifier can be written as:

$$L^{\mathcal{U}}_{style} = - \mathop{\mathbb{E}}_{\mathbf{x}_i \sim \mathcal{U}} \left[ \mathop{\mathbb{E}}_{\hat{\mathbf{x}}_i \sim p_D(\hat{\mathbf{x}}_i|E_s(\mathbf{x}_i), l_i)} \log p_C(l_i|\hat{\mathbf{x}}_i) + \mathop{\mathbb{E}}_{\tilde{\mathbf{x}}_i \sim p_D(\tilde{\mathbf{x}}_i|E_s(\mathbf{x}_i), \tilde{l}_i)} \log p_C(\tilde{l}_i|\tilde{\mathbf{x}}_i) \right]. \tag{10}$$

## 4 NEW DATASETS FOR CONTEXTUAL TEXT STYLE TRANSFER

Existing text style transfer datasets, no matter parallel or non-parallel, do not contain contextual information, thus are not suitable for our new task. Therefore, we introduce two new datasets: Enron-Context and Reddit-Context, derived from two existing datasets - Enron (Klimt & Yang, 2004) and Reddit Politics (Serban et al., 2017), respectively.

**Enron-Context** To build a formality transfer dataset with paragraph contexts, we randomly sampled emails from the Enron corpus (Klimt & Yang, 2004). After pre-processing and filtering with NLTK (Bird et al., 2009), we asked Amazon Mechanical Turk (AMT) annotators to identify informal sentences within each email, and rewrite them in a more formal style. Then, we asked different annotators to verify if each rewritten sentence is more formal than the original sentence.

**Reddit-Context** We further collected a new offensive vs. non-offensive dataset from the Reddit Politics corpus (Serban et al., 2017). First, we performed classification on the original dataset at sentence level, to identify offensive sentences from whole paragraphs. After filtering some extremely long/short sentences, we asked AMT annotators to rewrite the offensive ones to non-offensive alternatives. To provide robust datasets for benchmark, we randomly selected a subset of sentences to be rewritten into two references, which makes up of 10% of the whole dataset.

After manually removing wrong or duplicated annotations, we obtained a total of 14,734 rewritten sentences for Enron-Context, and 23,158 for Reddit-Context. We also limited the vocabulary size by using words with frequency equal or larger than 20 (70) in Enron (Reddit). Table 1 provides the statistics on the two datasets.

**Non-parallel Corpus** Besides parallel datasets, we also explore non-parallel datasets to enhance model training. For formality transfer, one choice is Grammarlys Yahoo Answers Formality Corpus (GYAFC) (Rao & Tetreault, 2018), crawled and annotated from two domains in Yahoo Answers.

| Formality Transfer | | | | | | | |
|---|---|---|---|---|---|---|---|
| Non-parallel | Train | Style classifier ‖ | Parallel | Train | Dev | Test | Coherence classifier |
| GYAFC | 58k | 12k ‖ | Enron-Context | 13k | 0.5k | 1k | 2.5k |

| Offensiveness Transfer | | | | | | | |
|---|---|---|---|---|---|---|---|
| Non-parallel | Train | Style classifier ‖ | Parallel | Train | Dev | Test | Coherence classifier |
| REDDIT | 106k | 15k ‖ | Reddit-Context | 22k | 0.5k | 1k | 3.5k |

Table 2: Statistics of the parallel and non-parallel datasets on the two text style transfer tasks.

This corpus contains paired informal and formal sentences, without context. We randomly selected a subset of sentences from the original dataset, and used it in a non-parallel manner. By the end, we collected 28,375/29,774 formal/informal sentences. The second dataset is the offensive/non-offensive Reddit dataset. Following dos Santos et al. (2018), we used a pre-trained classifier to extract offensive/non-offensive sentences from Reddit posts. In total, we collected 53,028/53,714 offensive/non-offensive sentences.

## 5 EXPERIMENTS

In this section, we compare our model with state-of-the-art baselines on the two new datasets, and provide both quantitative analysis and human evaluation to validate the effectiveness of our model.

### 5.1 EXPERIMENTAL SETUP

**Datasets** Table 2 provides a summary of the parallel and non-parallel datasets used for the two style transfer tasks. For the non-parallel datasets, we split them into two: one for the proposed model training, and the other for the style classifier pre-training. Similarly, for the parallel datasets, the training sets are divided into two as well, for the training of CAST and the coherence classifier, respectively.

**Evaluation Metrics** The contextual style transfer task requires generating sentences to: ($i$) preserve the original content and structure in the source sentence; ($ii$) conform to the pre-specified style; and ($iii$) align with the surrounding context in the paragraph. Thus, we consider the following automatic metrics to evaluate the effectiveness of different methods:

($i$) *Content Preservation.* We assess the degree of content preservation based on $n$-gram statistics, by measuring *BLEU* scores (Papineni et al., 2002) between generated sentences and human references. Following Rao & Tetreault (2018), we also use *GLEU* for the formality transfer task, which was originally introduced for the grammatical error correction task (Napoles et al., 2015). For offensiveness transfer, we include perplexity (*PPL*) as used in dos Santos et al. (2018), which is computed by a word-level LSTM language model pre-trained on non-offensive sentences.

($ii$) *Style Accuracy.* Similar to prior work, we generate samples from the model, and measure style accuracy (i.e., *Acc.* of the pre-trained style classifier).

($iii$) *Context Coherence.* As aforementioned, we use the pre-trained coherence classifier to measure how the generated sentences match the surrounding context.

For formality transfer, the pre-trained style classifier and coherence classifier reaches 91.35% and 86.78% accuracy, respectively. For offensiveness transfer, the accuracies are 93.47% and 84.96%, respectively. Therefore, we consider them as reliable to serve as evaluation metrics.

**Baselines** We compare our proposed model with several baselines: ($i$) Seq2Seq: a Transformer-based Seq2Seq model (i.e., Eqn. (2)), taking only sentences as inputs, and trained only on parallel data; ($ii$) Contextual Seq2Seq: a Transformer-based contextual Seq2Seq model (i.e., Eqn. (3)), taking both context and the sentence as input, and trained only on parallel data; ($iii$) Hybrid Seq2Seq (Xu et al., 2019): a Seq2Seq model leveraging both parallel and non-parallel data; ($iv$) ControlGen (Hu et al., 2017; 2018): a state-of-the-art text transfer model using non-parallel data.

**Implementation Details** The context encoder, sentence encoder and sentence decoder are all implemented as a one-layer Transformer with 4 heads. The hidden dimension of one head is 256, and

| | Formality Transfer | | | | Offensiveness Transfer | | | |
|---|---|---|---|---|---|---|---|---|
| Model | Acc. | Coherence | BLEU | GLEU | Acc. | Coherence | BLEU | PPL |
| Seq2Seq | 64.05 | 78.09 | 24.16 | 10.46 | 83.05 | 80.28 | 17.22 | 140.39 |
| Contextual Seq2Seq | 64.28 | 81.25 | 23.72 | 10.37 | 83.42 | 81.69 | 18.74 | 138.42 |
| Hybrid Seq2Seq | 65.09 | 79.62 | 24.35 | 10.93 | 83.28 | 84.87 | 20.78 | 107.12 |
| ControlGen | 62.18 | 73.66 | 14.32 | 8.72 | 82.15 | 78.81 | 10.44 | **92.14** |
| CAST | **68.04** | **85.47** | **26.38** | **15.06** | **88.45** | **85.98** | **23.92** | 93.03 |

Table 3: Quantitative evaluation results of different models on the two style transfer tasks .

| | | Task: informal to formal transfer | Context |
|---|---|---|---|
| A | Input
ControlGen
C-Seq2Seq
H-Seq2Seq
CAST | I'm assuming that you'd set up be part of that meeting ?
I'm guessing that you would be set up that call ?
I am assuming that you would part of that person .
I am assuming that you would be part of that party ?
Am I correct to assume that you would attend that meeting ? | I'll call him back to a meeting. [Input]. I asked him what sort of deals they're working on . |
| B | Input
ControlGen
C-Seq2Seq
H-Seq2Seq
CAST | Do y'all interface with C/P .
Do you compete with them ?
Do we interface with them ?
Do we interface with them ?
Do you all interface with C/P ? | Thanks . Can someone let the C/P know that the deals are good ? [Input]. If not deal confirmations could but they need the deal details . |
| | | **Task: offensive to non-offensive transfer** | **Context** |
| C | Input
ControlGen
C-Seq2Seq
H-Seq2Seq
CAST | You are ugly .
You bad guy !
Have a bad day .
What a bad day !
You look not good . | With the glasses , [Input]. I don't need them because I never read . How do i look ? |

Table 4: Examples from the two datasets, where orange denotes the sentence to be transferred, and blue denotes content that also appears in the context. **C-Seq2Seq**: Contextual Seq2Seq; **H-Seq2Seq**: Hybrid Seq2Seq.

the hidden dimension of the feed-forward sub-layer is 1024. The context encoder is set to take maximum of 50 words from the surrounding context of the target sentence. For the style classifier, we use a standard CNN-based sentence classifier (Kim, 2014).

Since the non-parallel corpus $\mathcal{U}$ contains more samples than the parallel corpus $\mathcal{P}$, we down-sample $\mathcal{U}$ to assign each mini-batch the same number of parallel and non-parallel samples to balance the training, alleviating the catastrophic forgetting problem described in Howard & Ruder (2018). We train the model using Adam optimizer with mini-batch size 64 and learning rate 0.0005. The validation set is used to select the best hyper-parameters. Hard-sampling (Logeswaran et al., 2018) is used to back-propagate the loss through discrete tokens from the pre-trained classifier to the model.

For the ControlGen (Hu et al., 2017) baseline, we use the code provided by the authors, and use their default hyper-parameter setting. For Hybrid Seq2Seq (Xu et al., 2019), we re-implement their model following the original paper.

## 5.2 EXPERIMENTAL RESULTS

**Formality Transfer** Results on the formality transfer task are summarized in Table 3. The CAST model, which leverages both context and non-parallel data, achieves the best performance over all the baselines. Particularly, CAST is able to boost *GLEU* and *Coherence* scores with a large margin. Hybrid Seq2Seq also achieves good performance by utilizing non-parallel data. By incorporating the context information, Contextual Seq2Seq also improves over the vanilla Seq2Seq model. As expected, ControlGen does not perform well, since only non-parallel data is used for training.

**Offensiveness Transfer** Results are summarized in Table 3. CAST achieves the best performance over all the metrics except for the *PPL*. In terms of **Coherence**, both methods that leverage the context achieve a better performance compared with the Seq2Seq baseline. Contextual Seq2Seq also improves *BLEU*, which is different from the observation in the formality transfer task. Our model produces slightly worse performance on *PPL* than ControlGen. We hypothesize that this is because our model tends to use the same non-offensive word to replace an offensive word, producing some unusual sentences.

| | Formality Transfer | | | | Offensiveness Transfer | | | |
|---|---|---|---|---|---|---|---|---|
| Model | *Acc.* | *Coherence* | *BLEU* | *GLEU* | *Acc.* | *Coherence* | *BLEU* | *PPL* |
| CAST | **68.04** | **85.47** | **26.38** | **15.06** | **88.45** | **85.98** | **23.92** | **93.03** |
| w/o context encoder | 65.35 | 82.9 | 23.98 | 14.17 | 84.15 | 80.96 | 20.54 | 127.02 |
| w/o cohere. classifier | 65.47 | 80.16 | 14.82 | 14.45 | 85.11 | 79.37 | 21.97 | 115.57 |
| w/o both | 62.19 | 74.47 | 15.88 | 10.46 | 72.69 | 78.15 | 13.14 | 147.31 |
| w/o non-parallel data | 60.19 | 75.49 | 13.5 | 9.88 | 70.84 | 78.72 | 10.53 | 151.08 |

Table 5: Ablation study of CAST on the two style transfer tasks.

| Task | Aspects | CAST vs. Contextual Seq2Seq | | | CAST vs. Hybrid Seq2Seq | | | CAST vs. ControlGen | | |
|---|---|---|---|---|---|---|---|---|---|---|
| | | win | lose | tie | win | lose | tie | win | lose | tie |
| Formality Transfer | Style Control | **57.1** | 28.3 | 14.6 | **46.9** | 26.1 | 28.0 | **72.1** | 12.6 | 25.3 |
| | Content Preservation | **59.7** | 22.1 | 18.2 | **50.4** | 20.8 | 28.2 | **68.8** | 14.5 | 17.7 |
| | Context Consistence | **56.4** | 23.1 | 20.5 | **51.5** | 19.7 | 28.8 | **70.1** | 10.6 | 19.3 |
| Offensiveness Transfer | Style Control | **58.6** | 25.3 | 16.1 | **50.1** | 29.2 | 20.3 | **54.8** | 19.9 | 25.3 |
| | Content Preservation | **62.3** | 26.5 | 11.2 | **54.0** | 17.5 | 28.5 | **53.1** | 30.2 | 16.7 |
| | Context Consistence | **60.1** | 32.4 | 17.5 | **55.3** | 24.9 | 20.8 | **58.1** | 35.8 | 16.7 |

Table 6: Results of pairwise human evaluation between CAST and three baselines on both tasks. Win/lose/tie indicate the percentage of texts generated by CAST are better/worse/equal to the compared model.

**Qualitative Examples** Table 4 presents some qualitative examples. Generally, we observe that our model is better at replacing informal words with formal ones (Example B and C), and generate more context-aware sentences (Example A and C), possibly due to the use of coherence and style classifiers. We also observe that the exploitation of the context information can also help our model preserve the semantic content in the original sentence (Example B).

**Ablation Study** To investigate the effectiveness of individual components, we perform ablation studies by removing some components of the proposed model. Results on both tasks are provided in Table 5. The context encoder and the coherence classifier play an important role in the proposed model. The context encoder is able to improve content preservation and style transfer accuracy, while the coherence classifier can help improve the coherence score but not much for style accuracy. By using these two components, our model can find a proper balance between transferring to the correct style and maintaining the consistency with context. When both of them are removed (the 4th row), performance on all the metrics drops significantly. We also observe that without using non-parallel data, the model performs poorly, showing the importance of using a hybrid method.

**Human Evaluation** Considering the subjective nature of this task, we conduct human evaluations based on content preservation, style control and context consistency, following Mir et al. (2019). Given the original sentence and the transferred sentence with the corresponding context, the AMT crowd-source workers were asked to select the best one based on these three aspects. The evaluation interface also allows a neutral option, if the worker considers both sentences as equally good in certain aspect. We randomly sampled 200 sentences from the corresponding test set, and collected three human responses for each pair. Table 6 reports the pairwise comparison results on both tasks. Based on human judgment, the quality of transferred sentences by CAST is significantly higher than the other methods on all three metrics. This is consistent with our observation in the experiments on automatic metrics.

## 6 CONCLUSION

In this paper, we present a new task - contextual text style transfer. To provide benchmarks for this new task, we introduce two new datasets, which contain annotated sentence pairs accompanied by paragraph contexts. We also propose a new model, which can jointly capture content preservation and context coherence, and exploit additional abundant non-parallel data for boosting performance. In both quantitative and human evaluations, our approach significantly outperforms baseline methods that do not rely on context information. Ablation study also demonstrates the effectiveness of different components in the model design. We believe our current model takes a first step towards modeling context information for text style transfer, and would like to explore more advanced solutions to integrating context.

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

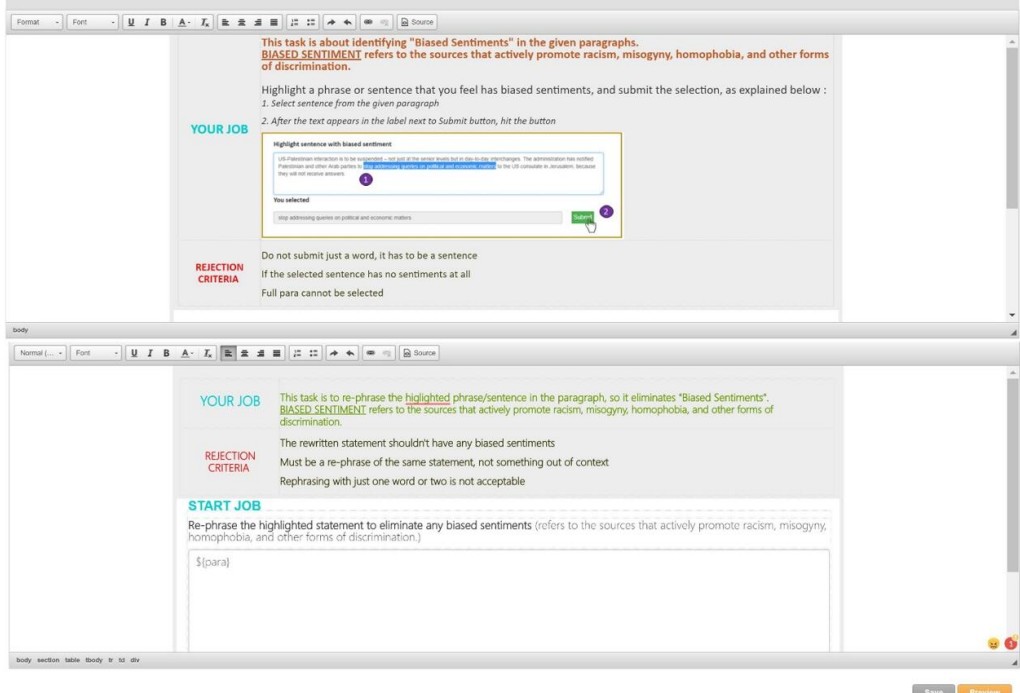

Figure 2: The AMT interfaces we used to collect the bias/violence data. The top interface is for turkers to select the bias/violence data from a paragraph. The bottom interface is for turkers to rewrite the sentence/phrase given the context.

**Data Collection:** We use the offensive language and hate speech classifier from (Davidson et al., 2017) to classify the offensive sentence. We perform the classification at the sentence level for each Reddit post.

As shown in Figure 2, the data collection are divided into two sub-tasks: first select the bias/violence data from a paragraph and then rewrite the sentence/phrase given the context.

The rewritten sentences from Enron-Context are validated by one of collaborators from a company. The 23,158 Reddit-Context are validated ourselves. Each rewritten sentence is reviewed by one volunteer to check if it is inoffensive while preserves the original content.

**AMT Interface:** In Figure 2, we show the AMT user interfaces to collect the bias/violence data.

**User Study:** In Table 7, we present more qualitative examples on both tasks.

| | | Task: informal to formal transfer | Context |
|---|---|---|---|
| A | **Input** | I'll share what I come up with later on . | If anyone has some of their classics, please forward them to me with a full citation. [Input]. - Bob |
| | **ControlGen** | i will share you what I come up later on . | |
| | **C-Seq2Seq** | I will call you later . | |
| | **H-Seq2Seq** | I will share with you later . | |
| | **CAST** | I will share my ideas later . | |
| B | **Input** | you were pissed last night . | You still mad at me? [Input]. Sorry. |
| | **ControlGen** | you were born last night . | |
| | **C-Seq2Seq** | we were upset last night . | |
| | **H-Seq2Seq** | you were upset last night . | |
| | **CAST** | You were angry last night . | |

| | | Task: offensive to non-offensive transfer | Context |
|---|---|---|---|
| C | **Input** | My f**king f**king tooth ! | Oh, it's my tooth! [Input]. I can't even think anymore because of it . |
| | **ControlGen** | My bad job ! | |
| | **C-Seq2Seq** | What a big big tooth . | |
| | **H-Seq2Seq** | What a bad news . | |
| | **CAST** | My broken tooth ! | |

Table 7: Additional examples from the two datasets, where orange denotes the sentence to be transferred, and blue denotes content that also appears in the context. **C-Seq2Seq**: Contextual Seq2Seq; **H-Seq2Seq**: Hybrid Seq2Seq.

