# OpenReview forum: "Contextual Text Style Transfer"
_ICLR.cc/2020/Conference — Reject_

### Official Review · AnonReviewer2 · 2019-10-20
**Official Blind Review #2**

**Rating:** 6

**Review:**


=== Summary ===

The paper considers the following task: given a sentence, its context paragraph, and the target style (formalness, inoffensiveness), generate a new sentence that still fits the original context. The evaluation metrics include content preservation (BLEU, GLEU, PPL), style accuracy (based on a pre-trained model), and context coherence (ditto).

The input context is the main highlight of this new task. The paper proposes a model that can use both parallel and non-parallel data. In addition to the existing non-parallel corpora, two datasets with crowdsourced parallel sentences were collected. Compared to the baselines, the proposed model produces sentences that are more relevant to the context.

=== Review ===

The major contributions of this paper are the new task setting and the accompanying parallel corpora. The datasets are rather large, and the fact that they contain parallel sentences (e.g., informal - formal sentences with the same semantics) makes the dataset potentially useful for other scenarios as well.

The proposed method is a natural extension of previous style transfer methods. One new addition is the contextual coherence loss, which is a relatively straightforward application of BERT-based models. If I am not missing anything, the model and the experiments are likely correct. In addition to the surrogate metrics, the paper also presents human evaluation, which strengthens the results.

**Experience Assessment:**

I do not know much about this area.

**Review Assessment: Checking Correctness Of Derivations And Theory:**

I assessed the sensibility of the derivations and theory.

**Review Assessment: Checking Correctness Of Experiments:**

I assessed the sensibility of the experiments.

**Review Assessment: Thoroughness In Paper Reading:**

I read the paper at least twice and used my best judgement in assessing the paper.

---

### Official Review · AnonReviewer3 · 2019-10-20
**Official Blind Review #3**

**Rating:** 3

**Review:**

The authors propose the task of contextual text style transfer: transferring the style of one text into another (i.e., informal to formal, or offensive to non-offensive), when the text is present within some larger, provided context. The authors propose a model (CAST) which takes advantage of the additional context to perform the style transfer. CAST outperforms previous style transfer models according to several automatic metrics, as well as human evaluation.

Overall, I am not convinced that the context is useful in the proposed tasks (Reddit-context and Enron-context), for 2 reasons:
* In the examples shown in Table 4, it seems possible to complete the proposed contextual style transfer tasks without the additional context (i.e., "Do y’all interface with C/P" -> "Do you all interface with C/P" for formal->informal transfer).
* CAST performs better across several metrics (including human evaluation), but I am not sure that the gain is from using the context in the manner that the authors suggest (increasing the generated sentence's coherence with the context). In particular, it is hard for me to see how some style transferred sentences can appear much more coherent than others, given that the the required changes are pretty local (i.e., word substitution). On the other hand, style transfer models seem to have some difficulty in maintaining semantic context, so it seems that some of the gains in e.g. BLEU could be from CAST using the context simply as a bag of words to draw from in generating a style-transferred sentence. Instead of including context, the issue of semantic correctness could be fixed in other, simpler ways, however, such as making a more extractive single-sentence style transfer model. I think some further analysis is necessary to conclude that the context makes the generations more coherent, rather than helping the model in some other way.

Other general concerns/questions for the authors:
* The baselines do not appear to have learned even the single-sentence style transfer task that well. It appears that the authors used the default hyper parameters for other style transfer models (Hybrid Seq2Seq and ControlGen), but wouldn't it be better to choose hyper parameters for those models based on the evaluation criteria (as is done for CAST)? I know that some style transfer models can be quite sensitive to hyperparameters, so I would be surprised if the models worked optimally out of the box. I also feel that a strong unsupervised machine translation model like XLM (or the related style transfer model from "Multiple-attribute text style transfer") could do better than the provided baselines, as these models seem to work notably better than their predecessors.
* I have a general concern that CAST is quite complicated and has a lot of moving parts and parameters to tune. I don't think there are ablations on all components of the model (i.e., the style classifier). I also wouldn't be surprised if a simple state-of-the-art unsupervised or semi-supervised machine translation model like XLM outperformed CAST, without using any context. To add the use of the context, it seems possible to make a simple modification to an existing state-of-the-art transfer/translation model like XLM (adding the context to the input)
* Why are the style and coherence classifiers fixed during training? Wouldn't it be better to update the classifiers while training the rest of the model (so they doesn't get exploited)?
* Which of the evaluated models/baselines are trained using the style and/or coherence classifiers directly used for evaluation? For models that are not trained to optimize the style and/or coherence classifier predictions, I am not sure it is fair to compare those models to CAST on the style/coherence classifier accuracy, since CAST has been directly trained to perform well on that metric.
* Are the evaluation classifiers identical to those used to train CAST? At the least, it seems better to use a different style/coherence classifier (i.e., trained with a different random seed or other hyperparameters) at test time that the one used during training (i.e., in case CAST has overfit the style/coherence of generations to the classifiers used during training).
* In Eqn. 5, why is there a tanh activation immediately preceding the softmax? Wouldn't the tanh limit the range of the output softmax probabilities?

The proposed datasets do seem useful to the community, and the empirical results do support that CAST performs better than baselines - it would just be helpful to understand in what way the gains are coming from the additional context (to know if the task is an interesting task).

**Experience Assessment:**

I have read many papers in this area.

**Review Assessment: Checking Correctness Of Derivations And Theory:**

N/A

**Review Assessment: Checking Correctness Of Experiments:**

I assessed the sensibility of the experiments.

**Review Assessment: Thoroughness In Paper Reading:**

I read the paper thoroughly.

---

### Official Review · AnonReviewer1 · 2019-10-24
**Official Blind Review #1**

**Rating:** 3

**Review:**

The paper proposes a new task for text style transfer, based on the idea that the the surrounding context of a sentence is important, whereas previous such tasks have only looked at sentences in isolation. Two new crowd-sourced datasets are created, and a combination of now fairly standard neural components is shown to outperform some strong baselines on the new datasets, on a variety of evaluation metrics. An ablation analysis of the components - including some auto-encoding auxiliary losses  - shows that all the various parts are helpful to performance.

Overall I don't see a lot wrong with the technical content of the paper, and it is well-written and easy to understand. I also feel that the collection of the data is a contribution, and this could be a useful dataset to the community. But the main problems I have with the current version - and the reason I am unable to recommend acceptance as it stands - are twofold:

1) the task doesn't make a lot of sense to me. If I have a paragraph written in a particular style, e.g. Shakespearean (an example in the paper), why would I want to rewrite a single sentence in e.g. modern English? Likewise, if I have a paragraph that's largely negative in sentiment (assuming for now this is a kind of style; see below), why would I want to change one of the sentences to a positive sentiment? I can understand why I might want to take a whole paragraph, or document, and rewrite it in another style, but not a sentence in isolation. Now it may be that the proposed method could be used consecutively to rewrite a larger unit, but it's not obvious how that would work and it needs some discussion.

2) it's not clear to me what is meant by "style" in this paper. No definition is provided, and the authors rely on references to previous papers and various examples throughout the paper for the reader to grasp what is meant here. Examples include: informal/formal, offensive/inoffensive, Shakespearean, sentiment, political slant. The first three look fine as examples of style; the latter two I'm not sure, since they are likely to change the semantic content. A minor point related to this one is that in the Text Style Transfer related work subsection, there's even a reference to an MT paper as an example.

Having said all that, I think the new corpus could potentially be a useful resource (assuming the authors can utilise it for a meaningful task). I would have liked to see more details of how the corpus was collected. Putting the style guidelines in an Appendix would be a possibility. But more details could be given in section 4, for example: how are the offensive sentences identified? (which classifier?) How did you manually remove wrong annotations, and what does wrong mean here? (presumably not by checking all 14k examples?)


**Experience Assessment:**

I have read many papers in this area.

**Review Assessment: Checking Correctness Of Derivations And Theory:**

I assessed the sensibility of the derivations and theory.

**Review Assessment: Checking Correctness Of Experiments:**

I assessed the sensibility of the experiments.

**Review Assessment: Thoroughness In Paper Reading:**

I read the paper at least twice and used my best judgement in assessing the paper.

---

### Decision · Program_Chairs · 2019-12-19

**Decision:**

Reject

**Comment:**

The paper proposes a new style transfer task, contextual style transfer, which hypothesises that the document context of the sentence is important, as opposed to previous work which only looked at sentence context. A major contribution of the paper is the creation of two new crowd-sourced datasets, Enron-Context and Reddit-Context focussed on formality and offensiveness. The reviewers are skeptical that it was context that has really improved results on the style transfer tasks. The authors responded to all the reviewers but there was no further discussion. I feel like this paper has not convinced me or the reviewers of the strength of its contribution and, although interesting, I recommend for it to be rejected.